# Complement Inhibitors Vitronectin and Clusterin Are Recruited from Human Serum to the Surface of Coronavirus OC43-Infected Lung Cells through Antibody-Dependent Mechanisms

**DOI:** 10.3390/v14010029

**Published:** 2021-12-24

**Authors:** Candace R. Fox, Griffith D. Parks

**Affiliations:** Burnett School of Biomedical Sciences, College of Medicine, University of Central Florida, Orlando, FL 32827, USA; griffith.parks@ucf.edu

**Keywords:** coronavirus, parainfluenza virus, complement

## Abstract

Little is known about the role of complement (C’) in infections with highly prevalent circulating human coronaviruses such as OC43, a group of viruses of major public health concern. Treatment of OC43-infected human lung cells with human serum resulted in C3 deposition on their surfaces and generation of C5a, indicating robust C’ activation. Real-time cell viability assays showed that in vitro C’-mediated lysis of OC43 infected cells requires C3, C5 and C6 but not C7, and was substantially delayed as compared to rapid C’-mediated killing of parainfluenza virus type 5 (PIV5)-infected cells. In cells co-infected with OC43 and PIV5, C’-mediated lysis was delayed, similar to OC43 infected cells alone, suggesting that OC43 infection induced dominant inhibitory signals. When OC43-infected cells were treated with human serum, their cell surfaces contained both Vitronectin (VN) and Clusterin (CLU), two host cell C’ inhibitors that can alter membrane attack complex (MAC) formation and C’-mediated killing. VN and CLU were not bound to OC43-infected cells after treatment with antibody-depleted serum. Reconstitution experiments with purified IgG and VN showed that human antibodies are both necessary and sufficient for VN recruitment to OC43-infected lung cells–novel findings with implications for CoV pathogenesis.

## 1. Introduction

Human respiratory viruses are a major public health concern and impose a huge burden on the economy and the health care industry. Non-influenza related respiratory virus infections account for nearly 40 billion dollars annually in direct and indirect medical costs in the United States alone, with similar costs for chronic conditions such as hypertension and congestive heart failure [1]. Pathogens associated with these illnesses include enveloped RNA viruses, such as coronaviruses, which remain highly prevalent in the human population, with reoccurring seasonal infections. Infections with common circulating coronaviruses, such as strain OC43, tend to surge, starting in the winter and prolonged until spring [2]. There is an urgent need for better therapeutic approaches to prevent the transmission and diseases associated with these respiratory viruses.

A key pathway of the innate immune responses is the complement (C’) system, which most animal viruses must encounter during infections. Soluble and cell membrane-associated proteins coordinate C’-mediated defenses against viral infections. This can include multiple mechanisms such as direct virus recognition and neutralization, B and T cell activation, leukocyte recruitment and stimulation, and virus opsonization by immune cells [3,4,5]. Consequentially, viruses have evolved strategies to prevent C’ pathway activation and execution, which may contribute to viral pathogenesis and disease (e.g., [6,7,8,9]). Virus-C’ pathway interactions need to be fully elucidated for developing more effective vaccines and therapeutic vectors [10,11,12].

C’ is activated through virus structure recognition by the classical, lectin, or alternative pathways, which then converge on a central component C3 [13,14]. C3 is cleaved into C3a, an anaphylatoxin to increase inflammation, and C3b. Opsonization and phagocytosis is enhanced by C3b covalently binding to viral components. C3b can additionally associate with other factors to form the C3 convertase (e.g., C3bBb), and can amplify the initially deposited C3b by further cleavage of C3 in a feedback loop [15]. Further downstream activation of factors such as C5 through C9 can lead to formation of the membrane attack complex (MAC), which is capable of lysing virus particles or infected cells.

The C’ system is highly regulated to prevent inappropriate damage to normal cells and healthy tissues (e.g., [16,17]), which involves a series of host cell C’ activation regulators and C’ inhibitors. MAC formation can be inhibited by soluble factors found in serum, such as vitronectin (VN) and clusterin (CLU). Both VN and CLU prevent efficient MAC insertion into host cell membranes and inhibit C’-mediated lysis of host cells. VN blocks the C5b-7 complex from binding to lipids, whereas CLU prevents C5b-7 insertion into virions or host cell membranes.

Many large DNA viruses encode analogs that directly inhibit C’ pathways or act as mimics of host cell regulators are reviewed in [18,19,20]. By contrast, the small coding capacity of most RNA virus genomes is thought to drive these viruses to associate with soluble or membrane-bound host cell regulators as a mechanism to limit C’ neutralization [9,21]. Examples of the recruitment of host C’ inhibitors include human immunodeficiency virus type 1 (HIV 1) which incorporates CD55, CD59 and CD46 into progeny virions [22], and hepatitis C virus which assembles with CD55 [23]. Parainfluenza virus type 5 (PIV5) recruits cellular C’ inhibitors CD55, CD46 and CD59 during budding [24,25,26], and PIV5 infection upregulates the synthesis of CD55 to produce virions with enhanced resistance to neutralization by C’ [27].

Given that our understanding of CoVs interactions with C’ pathways is incomplete, we have tested the capacity of C’ to lyse human lung epithelial cells infected with the circulating coronavirus strain OC43. Using a novel real-time assay for C’-mediated cell killing, we show that treatment of OC43-infected cells results in C3 deposition on their surfaces. However, lysis of infected cells is very slow relative to that of PIV5-infected cells. OC43 infection results in the recruitment of two C’ inhibitors, vitronectin (VN) and clusterin (CLU), from serum to the cell surface in an antibody-dependent mechanism.

## 2. Materials and Methods

### 2.1. Cells, Viruses, and Infections

Annette Khaled (University of Central Florida, Orlando, FL, USA) kindly supplied the H1299 non-small cell lung carcinoma cell line. Robert Lamb (Northwestern University, Evanston, IL, USA) provided the Vero and CV-1 cell lines. H1299 cells and HCT-8 (ATCC, Manassas, VA, USA) cells were grown in 10% heat inactivated fetal calf serum (HI FBS, Gibco, Thermo Fisher Scientific, Waltham, MA, USA) Roswell Park Memorial Institute medium (RPMI 1640). H1299 cells expressing a nuclear red fluorescent protein (H1299-NLR cells) were generated by transduction with NucLight Red lentivirus (Sartorius, Göttingen, Germany) followed by 1 μg/mL puromycin selection. Cultures of Vero, RD, and CV-1 cells were grown in Dulbecco modified Eagle medium (DMEM) supplemented with 10% HI FBS. Infections were performed by incubating virus and cells in RPMI supplemented with 2% HI FBS.

Human Coronavirus OC43 (ATCC, catalog number VR-1558) was grown in HCT-8 cells at 33 °C. Media collected at 4 days post infection (dpi) was clarified by centrifugation and then a 1:20 dilution of Bovine Albumin Fraction V (BSA, 7.5% solution, Fisher) was added to the solution. Aliquots were quick frozen using liquid nitrogen and stored at −80 °C. 50% Tissue Culture Infectious Dose assays (TCID_50_) on confluent RD cells in 96-well plates (Falcon, Thermo Fisher Scientific) were performed to quantify OC43 stocks. Briefly, solutions were serially diluted in DMEM containing a 1:20 dilution of BSA. After incubation with diluted virus for one hour (hr) at 33 °C, cells were washed and replaced with DMEM containing 2% HI FBS. After 4 days of incubation at 33 °C, cells were washed and stained with a 1% crystal violet solution containing 20% ethanol, 3.7% formaldehyde and PBS for 30 min. TCID_50_ values were calculated by the Spearman and Kärber algorithm as previously described [28]. The parainfluenza virus 5 leader mutant (PIV5) expressing GFP was grown in Vero cells and titered on CV-1 cells as previously described [29]. For OC43 infections, H1299 lung cells were infected at a multiplicity of infection (MOI) of 5 TCID_50_ units per cell in 2% HI FBS RPMI and processed at 2 dpi, unless otherwise denoted. For PIV5 infections, H1299 lung cells were infected at a MOI of five plaque forming units (PFU) units per cell and processed at 17 h post infection (hpi), unless otherwise denoted.

### 2.2. Flow Cytometry

Cells cultured in 24-well plates (2 cm diameter) were infected with OC43 and harvested at various dpi. Media and trypsinized adherent cells were centrifuged, fixed and permeabilized using Invitrogen eBioscience (Thermo Fisher Scientific) reagents according to manufacturer’s recommendations. Samples were then stained for OC43 protein expression using a primary antibody targeted toward OC43 nucleocapsid protein (NP, MAB9013, Sigma-Aldrich, St. Louis, MO, USA) followed by secondary AlexaFluor 488 (Invitrogen, Thermo Fisher Scientific) antibody staining. Cells were quantified by flow cytometry using the CytoFLEX (Beckman Coulter, Brea, CA, USA) and 10,000 independent events were analyzed using CytExpert software (Beckman Coulter).

Cells cultured in 6-well plates (10 cm diameter) were infected with OC43 at an MOI of 5 TCID_50_ units per cell and harvested at 2 dpi. Normal human serum (NHS, Innovative Research, Novi, MI) was utilized and sera were heat inactivated (HI) by heating to 56 °C for 30 min. Reactions were prepared to include cells either untreated, HI NHS, or NHS treated in 10% HI FBS RPMI. Reactions were incubated for 1 h at 37 °C, followed by a media wash and centrifugation. Cells were then surface stained with an APC-conjugated antibody against IgG Fc (410601, Biolegend, San Diego, CA, USA), FITC-conjugated antibody against C3 (855167, MP Biomedicals, Irvine, CA, USA), SC5b-9 (A239, Quidel, San Diego, CA, USA), VN ([1G11E8] ab201981, Abcam, Cambridge, United Kingdom), and CLU (A241, Quidel). Cells were washed followed by appropriate secondary antibody staining if needed using AlexaFluor 488 and 405 antibodies (A31553, Invitrogen, Thermo Fisher Scientific).

As indicated in the figure legends, reactions included 200 μg/mL recombinant VN (757708, Biolegend), 7 mg/mL purified IgG (14506, Sigma), 10% C3, C5, C6, C7-depleted sera (Complement Technologies, Tyler, TX, USA), or 10% IgG, IgA, IgM and IgE antibody depleted serum (HS2001, Celprogen, Torrance, CA, USA).

### 2.3. Cell Viability Assays

Cells cultured in 24-well plates (2 cm diameter) were treated as indicated in the figure legends. After treatment, both media and trypsinized adherent cells were harvested and stained for propidium iodide (BD Bioscience) as described by the manufacturer. Samples were then processed for by flow cytometry on the CytoFLEX (Beckman Coulter, Brea, CA, USA) and 10,000 independent events were analyzed using CytExpert software (Beckman Coulter).

Cell viability assays were also performed using the IncuCyte instrument (Sartorius) as previously described [30]. Briefly, uninfected (mock) or OC43 infected H1299-NLR cells were plated in triplicate in 96-well plates (Corning) at 7000 cells/well. Plates were maintained in the IncuCyte CO2 incubator for 3–4 days (d), while images were captured every 2 h using 10× objective in red and phase channels. Red object count (ROC) per well was calculated and scans were normalized to the value at time zero (ROC^t0^) when treatments were added. Pooled normal human serum (NHS) was purchased (Innovative Research, Novi, MI, USA). Sera were heat inactivated (HI) by heating to 56 °C for 30 min. C3, C5, C6, C7-depleted sera and their respective purified recombinant proteins were purchased from Complement Technologies (Tyler, TX, USA).

### 2.4. Western Blotting

Six-well dishes (60 mm diameter) of cells were treated as described in the figure legends, followed by lysis in protein lysis buffer (Cell Signaling Technology, Danvers, MA, USA). Cell lysate was resolved on 12% sodium dodecyl sulfate-polyacrylamide gel electrophoresis (SDS-PAGE) gels (Bio-Rad, Hercules, CA, USA) and transferred to nitrocellulose membranes. Samples were normalized to β–actin (1:10,000 dilution, A5316, Sigma-Aldrich, St. Louis, MO, USA) expression and probed with antibodies for VN (1:1000 dilution, GTX103475, GeneTex, Irvine, CA, USA) and CLU (1:1000, A241, Quidel, San Diego, CA, USA). Blots were visualized by horseradish peroxidase-conjugated antibodies (Sigma-Aldrich) and chemiluminescence (Thermo Fisher Scientific).

### 2.5. Immunostaining

Cells were grown in 48-well plates and treated as indicated in the figure legends. C3 immunostaining was performed using an FITC conjugated antibody against C3 at a 1:1000 dilution (855167, MP Biomedicals, Irvine, CA, USA). Images were acquired with 20× objective lens on a Keyence All-in-One Fluorescence Microscope BZ-X800 (Keyence America, Itasca, IL, USA). Vitronectin immunostaining was performed using a primary VN antibody (1:100 dilution, GTX103475, GeneTex) followed by secondary anti-rabbit AlexaFluor 488 (1:1000 dilution, A11008. Invitrogen, Thermo Fisher Scientific). Images were acquired with 20× objective lens on the IncuCyte instrument (Sartorius).

### 2.6. Human C5a ELISA

Six-well dishes (60 mm diameter) of cells were treated as detailed in the figure legends and supernatants were evaluated using a Human C5a ELISA kit as described by the manufacturer (557965, BD Opt EIA; BD Biosciences, Franklin Lakes, NJ, USA). ELISA results were normalized to 10^6^ cells.

### 2.7. Statistical Analyses

Values are the mean of three replicates and experiments were performed at least twice. Statistical analysis was performed using Prism GraphPad, students T test or a two-way analysis of variance (ANOVA) using tukey’s post hoc test over the time course studies. In all figures, * indicates *p*-value < 0.05, ** indicates *p*-value < 0.01, and *** indicates *p*-value < 0.001.

## 3. Results

### 3.1. Treatment of OC43-Infected Human Lung Cells with NHS Results in Deposition of Antibodies and C3

To determine the kinetics of OC43 infection and cell killing, H1299 cells were either uninfected (mock) or infected with OC43 at an MOI of 5 TCID_50_/cell. At 24, 48, and 72 h post infection (hpi), cells were harvested and stained for either intracellular expression of OC43 nucleocapsid (NP) protein (Figure 1A) or for propidium iodide (PI) as a measure of cell death (Figure 1B). These timecourse studies demonstrated that OC43 productively infected ~70%–80% of the cells by 48 hpi with minimal cell death (i.e., PI staining).

To test if the OC43-specific antibodies contained in NHS can bind to OC43 infected lung cells, uninfected (mock) and OC43 infected cells were either untreated or treated with 1% or 5% NHS for 1 h at 37 °C. Cells were washed and stained for the presence of surface Fc protein. As shown in Figure 1C, uninfected (mock) cells showed minimal antibody staining, whereas OC43 infected cells treated with either 1% or 5% NHS resulted in 50% and 95% Fc staining, respectively, suggesting antibodies in NHS can recognize and bind to OC43 infected cells. Cells were also stained for C3 deposition at the cell surface. As shown in Figure 1D, ~80%–90% of OC43 infected cells treated with 5% NHS were found to stain positive for C3 deposition, indicating that the C’ pathway is activated by OC43 infected cells and components of the pathway can bind to infected cells.

To determine the extent to which NHS can kill OC43 infected cells, uninfected (mock) and OC43 infected cells were treated at 2 dpi with 10% NHS for either 4 h or 16 h, followed by PI staining to assay for cell death. HI NHS was included as an inactivated C’ control. Both mock and OC43 infected cells had basal levels of cell death around 20% PI positive when treated with NHS for 4 h (Figure 1E). By contrast, when cells were treated for 16 h with 10% NHS, uninfected (mock) cells stained about 25% PI positive, whereas ~60% of the OC43 infected cells were positive for PI staining (Figure 1F). Treatment with HI NHS resulted in a lower level of PI positive cells in the case of OC43 infection. Together, these data indicate that antibodies and C3 contained in NHS are deposited on the surface of OC43 infected cells, and this correlates with lysis of OC43 infected cells.

### 3.2. Real Time Cell Killing Assays Reveal That NHS Lyses OC43 Infected Lung Cells at a Slower Rate Than Parainfluenza Virus Infected Lung Cells and Require Complement C3

We utilized our previously described real-time cell monitoring assay [30] with an IncuCyte instrument to define the kinetics of C’-mediated killing of OC43 infected lung cells. Using H1299 cells that stably express a nuclear red fluorescent protein (H1299-NLR), the IncuCyte instrument can record both bright field and red fluorescence images at real-time continuous intervals. As such, H1299-NLR cells were uninfected (mock) or infected with OC43 at an MOI of 5 TCID_50_ units per cell and, at 48 hpi, cells were incubated alone or with 10% NHS or HI NHS. For each experiment, red-labeled cells were determined per well, normalized to the initial values, and expressed as a percentage of time zero when treatments were added.

Representative images are shown in Figure 2A,C,E and the percentage of red object count (ROC) normalized to time zero was quantified and is shown in Figure 2B,D,F. Uninfected (mock) cells that were untreated (blue curves) or treated with NHS or HI NHS (red and green curves) continued to proliferate, producing red intensity that increased to levels about 250% of time zero (Figure 2A,B). Untreated OC43 infected cells (blue curve) maintained roughly 100% of time zero suggesting the viral infection leads to a cytostatic environment in the cell population, which could be due to slower growth curves or a combination of cell death and proliferation that are at similar rates (Figure 2C,D). Most importantly, NHS-treated OC43 infected cells (green curve) showed a loss of red intensity over time, reaching ~50% of time zero after 30 h NHS treatment. This loss of cells did not occur following treatment with HI NHS (red curve).

The kinetics of NHS-mediated killing of OC43 infected cells was compared to PIV5-infected cells, since we have previously shown that PIV5 is neutralized by what appears to be mechanisms that are complement-dependent but specific antibody-independent [25,27,31,32]. As shown in Figure 2E,F, untreated and HI NHS-treated PIV5-infected cells (blue and red curves) showed slow increases in red fluorescence relative to time zero due in part to the low cytopathic effects of PIV5 infections [33]. By contrast, NHS-treated PIV5 infected cells (green curve) showed a very rapid reduction in intensity and in number of cells to ~50% after only 4 h NHS treatment. Taken together, these results from the real-time viability assays show that NHS treatment results in much slower lysis of OC43 infected cells (50% killing at ~25 h post treatment) as compared to PIV5 infected cells (50% killing at ~4 h post treatment).

C3 is a major component of the C’ cascade, and C3 cleavage is a critical step involved in activation and subsequent cell lysis [15]. A possible explanation for the delayed cell killing of OC43-infected cells is that C3 could be deposited on only a fraction of the OC43 infected cells leading to lysis of only sub-population of cells. To address this hypothesis, uninfected (mock) and OC43 infected cells were either untreated or treated with HI NHS or NHS for 1 h, stained for cell surface C3 deposition and quantified by flow cytometry (Figure 3A). Uninfected (mock) cells had minimal surface C3 staining in all treatment conditions. By contrast, >80% of the OC43 infected cells treated with NHS stained positive for C3. Staining for C3 was not detected when HI NHS was used. Additionally, NHS treated OC43-infected cells showed an approximately 5-fold mean fluorescence intensity (MFI) increase in C3 positive staining as compared to mock NHS treated cells, whereas PIV5-infected NHS treated cells demonstrated about 3-fold MFI increase (data not shown). These results suggested similar levels of C3 deposition and C’ activation following NHS treatment of both OC43- and PIV5-infected. This result was confirmed using cell surface C3 staining determined by immunofluorescence (Figure 3B). These results indicate that the slow killing of OC43-infected cells by NHS is not due to only a small population of cells having C3 deposition.

To determine the role of C3 in NHS-mediated cell lysis, uninfected (mock) or OC43 infected H1299-NLR cells were left untreated, or were treated with HI NHS, NHS, C3-depleted (depl) serum or C3-depl serum reconstituted with physiological levels of exogenously added C3. Cells were imaged at continuous intervals on the IncuCyte instrument and ROC were normalized and expressed as percentage of time zero. Uninfected (mock) cells proliferated to about 250% of time zero under all treatment conditions (data not shown). OC43 infected cells that were treated with C3-depl serum (Figure 3C, purple curve) had red fluorescence intensity curves that matched that of cells left untreated or treated with HI NHS, indicating that C3 was an important factor for NHS-mediated killing. Most importantly, cells treated with C3-depl serum which had been reconstituted with purified C3 demonstrated a gradual declining loss of red intensity similar to that seen with NHS-treated infected cells.

### 3.3. Complement-Mediate Lysis of OC43-Infected Cells Requires C5 and C6, but Not C7

Assembly of the terminal membrane attack complex (MAC) starts with C5 cleavage into C5a and C5b, and C5b can then interact with C6 and C7, generating a C5b-7 complex that can physically associate with target cell membranes. C5b-7 can then recruit C8 to form C5b-8 which acts as a receptor for C9 binding and polymerization, forming the lytic MAC pore [34]. To determine if C5 was activated in response to OC43 infected cells, uninfected (mock) or OC43 or PIV5 infected cells were either untreated or treated with NHS for 4 h. Supernatants were then collected and analyzed for C5a levels (Figure 4A). Supernatant from NHS treated OC43 infected cells yielded ~400 ng/mL of C5a, similar levels to PIV5 infected cells treated with NHS. These results suggest downstream C5 pathways are activated in response to both OC43 and PIV5 infected cells.

To determine if C’ activation results in the deposition of MAC components on OC43 infected cells, uninfected (mock) or OC43-infected cells were left untreated or treated with HI NHS or NHS for 1 h and then stained with an antibody which recognizes the C5b-9 complex. As shown in Figure 4B, uninfected (mock) cells treated with NHS showed minimal surface C5b-9 staining, whereas OC43 infected cells that were treated with NHS had about 70% of the population stained positive for C5b-9 deposition. These data indicate that at least some components of the MAC forms on the surface of OC43 infected cells when treated with NHS.

We determined which C’ terminal pathway proteins were required for C’-mediated lysis of OC43 infected cells. Uninfected (mock) or OC43 infected H1299-NLR cells were left untreated, or were treated with HI NHS, NHS, or with C5-, C6- or C7-depleted (depl) sera. In addition, controls were included of depl-sera that had been supplemented with physiological levels of the relevant purified proteins [34,35]. Cells were imaged at continuous intervals on the IncuCyte instrument, ROC were normalized and expressed as percentage of time zero. As shown in Figure 4C,E,G, uninfected (mock) cells proliferated to about 250% of time zero under all treatment conditions. By contrast, OC43-infected cells treated with C5- and C6-depl sera (Figure 4D,F, purple curves) aligned with untreated and HI NHS curves, suggesting C5 and C6 are required for C’-mediated lysis of OC43 infected cells. Consistent with this interpretation, when purified C5 or C6 were supplemented back to their corresponding depl-sera (Figure 4D,F, orange curves), the OC43 infected cells showed a gradual decline in cells similar to that seen with NHS treated cells. By contrast to results with C5 and C6, when OC43 infected cells were treated with C7-depl serum (Figure 4H, purple curve), there was a loss of cell number which was similar to that seen with NHS and addition of C7 to C7-depl serum did not alter the kinetics of cell loss (orange and green curves, respectively). Taken together, these data suggest that an MAC forms on OC43 infected cells when treated with NHS and these cells are targeted for cell lysis by a pathway that requires C5 and C6, but not C7.

### 3.4. OC43 Infected Lung Cells Actively Delay C’-Mediated Lysis

The above data lead to the hypothesis that OC43 infection results in active inhibition of C’-mediated cell lysis, which leads to delayed death. A prediction of this hypothesis is that the rapid C’-mediated killing of PIV5 infected cells would be slowed in the case of cells that are co-infected with OC43 and PIV5—due to the dominant inhibitory mechanism of OC43 infection. To test this prediction, H1299-NLR cells were infected with OC43 at an MOI of 5 TCID_50_ units per cell for 31 h, and were then subsequently co-infected with PIV5, which expresses GFP [29] at an MOI of 5 PFU per cell. After 17 additional hours, cells were then untreated or treated with NHS to identify co-infected cells that were positive for Fc staining and GFP expression by flow cytometry analysis as depicted on a timeline (Figure 5A). As shown in Figure 5B,C, both OC43 infected and OC43/PIV5 co-infected cells showed ~90% of the population staining positive for bound antibodies suggesting PIV5 did not interfere with levels of OC43 infection in the co-infected samples. Conversely, PIV5 infected cells treated with NHS had minimal antibody binding (green bar, Figure 5B). These results demonstrate that NHS does not contain antibodies that can recognize PIV5-infected H1299 cells. This is consistent with our published work [32] showing that human serum does not contain antibodies that bind to the surface of PIV5-infected A549 lung cells. The percentage of cells that were IgG positive did not change between cells infected with OC43 alone and cells co-infected with OC43 and PIV5. Since PIV5 infected cells do not bind IgG from NHS, we conclude that the percentage of OC43 infected cells in the co-infection did not change. PIV5-derived GFP expression ranged from 70%–90% in both PIV5 infected and OC43/PIV5 co-infected cells (Figure 5C). When the population was gated on OC43 Fc+ cells, GFP expression was about 80% positive (data not shown) suggesting OC43 infected cells can be productively co-infected with PIV5.

To test the prediction that OC43 infection would convert the rapid C’-mediated killing of PIV5 cells to the slow killing of OC43-infected cells, H1299-NLR cells were uninfected (mock) or infected with OC43 or PIV5 alone, or co-infected with OC43 and PIV5. Cells were left untreated or treated with 10% HI NHS, or 10% NHS and monitored on the IncuCyte instrument. As shown in Figure 5D–F, uninfected (mock) cells grew to 250% of time zero under all conditions, OC43-infected cells showed a reduction to ~50% of time zero by 30 h post treatment (Figure 5E, green curve), and PIV5 infected cells showed a reduction to 50% of time zero within only ~4 h of NHS treatment (Figure 5F, green curve). Most importantly for the hypothesis tested, cells that were co-infected with OC43 and PIV5 showed NHS-dependent kinetics of a loss to 50% of time zero by 30 h post treatment (Figure 5G, green curve), similar to that seen with cells infected with OC43 alone. Together, these data show that OC43 infected cells can delay C’-mediated lysis even when co-infected with a potent C’ activator. This supports the hypothesis that OC43 infection results in active inhibition of C’-mediated cells killing which results in delayed lysis.

### 3.5. NHS Treatment of OC43-Infected Human Lung Cells Results in Deposition of C’ Inhibitors VN and CLU

Our results indicate that C7 is not required for C’-mediated lysis of OC43 infected cells (Figure 4H) and that C’ inhibitors may be preventing cell lysis. VN and CLU are two cellular C’ inhibitors that alter the activity of the MAC to prevent lysis of target cells. To address the hypothesis OC43 infected cells can recruit these two C’ inhibitors from serum, uninfected (mock) or OC43 infected cells were either untreated, treated with 10% HI NHS or 10% NHS for 1 h. Cell lysates were probed with antibodies specific for VN and CLU. Both VN and CLU were detected in OC43 infected samples treated with HI NHS and NHS (Figure 6A, lanes 5 and 6, respectively).

Alternatively, washed and intact cells were stained for surface VN and CLU expression and analyzed by flow cytometry. Approximately, 50%–60% of OC43 infected cells treated with either HI NHS or NHS stained positive for surface VN expression (Figure 6B) and mean fluorescence intensity (MFI) reflected this increase in VN binding (Figure 6D). Similarly, ~40%–60% of OC43 infected cells treated with either HI NHS or NHS stained positive for surface CLU expression (Figure 6C) and MFI correlated with an increase in CLU binding (Figure 6E). Taken together, these data suggested that VN and CLU are recruited from serum to bind to the surface of OC43 infected cells.

Since OC43 co-infected with PIV5 cells display a delayed C’-mediated lysis as compared to PIV5 infected cells (Figure 5), we hypothesized that VN and CLU would be bound to OC43 co-infected with PIV5 to similar levels as OC43 infected cells. Approximately, 60–70% of both OC43 infected cells and OC43 co-infected with PIV5 treated with either HI NHS or NHS stained positive for surface VN expression (Figure 6F). OC43 and PIV5 coinfection also results similar levels of CLU recruitment (Figure 6G). Taken together, these results suggested that VN and CLU bind to the surface of cells infected with OC43 and OC43 plus PIV5 but not to cells infected with PIV5 alone.

To visualize VN binding from serum to OC43 infected cells, uninfected (mock) or OC43 infected H1299 NLR cells were either untreated or treated with 10% NHS, washed, followed by cell surface staining with an antibody specific for VN. As shown in Figure 7, NHS treatment of OC43 infected cells resulted in VN binding to the infected cell surface (Figure 7A), consistent with our flow cytometry data (Figure 6B). Image quantification revealed that uninfected (mock) NHS treated samples had about 100 VN positive cells per image whereas, strikingly, NHS treatment of OC43 infected cells resulted in approximately 800 VN positive cells per image (Figure 7B).

To determine if VN alone could recognize and bind to OC43 infected cells, uninfected (mock) or OC43 infected lung cells were either untreated, treated with 10% NHS, or treated with physiological levels of VN [36] for 1 h at 37 °C. Cells were washed and stained with antibodies for surface VN. As shown in Figure 8A, recombinant VN treatment alone was unable to bind to OC43 infected cells, suggesting other serum components are required for VN cell surface binding.

To determine what key C’ serum factors are required for VN recruitment to OC43 infected cells, uninfected (mock) or OC43 infected cells were either left untreated or were treated with NHS or sera depleted of C3, C5, C6, or C7 for 1 h at 37 °C. Cells were then washed and probed for surface VN (Figure 8B). Unexpectedly, OC43 infected cells treated with each of the depleted sera had high VN surface staining similar to that of the NHS treated infected cells.

Since VN required soluble factors from serum to bind to OC43 infected cells and key MAC pathway components were not required for VN binding, we tested the hypothesis that antibodies are required for VN and CLU binding. Uninfected (mock) or OC43-infected cells were untreated, or treated with NHS, antibody depleted serum alone, or antibody depleted serum plus purified IgG from human serum at physiological levels [37] for 1 h at 37 °C. Cells were then washed and immunostained for surface VN (Figure 8C) or CLU (Figure 8D). OC43 infected cells treated with antibody-depl serum had no detectable antibody binding and when the reaction was reconstituted with physiological levels of purified human IgG, OC43 infected cells had bound antibodies similar to that of NHS treatment (data not shown). OC43 infected cells treated with antibody depl serum had minimal VN and CLU binding, whereas when the reaction was reconstituted with purified human IgG, OC43 infected cells had bound surface VN and CLU staining similar to that of NHS treatment (Figure 8C,D). These data suggested that VN and CLU recruitment to OC43 infected cells required antibodies from human serum.

Since antibodies were necessary for VN to bind to OC43 infected cells, we next sought to determine if antibodies alone were sufficient for VN binding. Uninfected (mock) or OC43 infected cells were untreated or treated with NHS, physiological levels of recombinant vitronectin alone, or recombinant VN plus purified IgG for 1 h. Cells were then washed and immunostained for surface antibody Fc (Figure 8E) and surface VN (Figure 8F). OC43 infected cells treated with VN alone had no detectable antibody binding and when the reaction was reconstituted with physiological levels of purified human IgG, OC43 infected cells had bound antibodies similar to that of NHS treatment (Figure 8E). OC43 infected cells treated with recombinant VN had minimal VN binding, conversely, when the reaction was reconstituted with purified human IgG, ~50% of OC43 infected cells had bound surface VN staining (Figure 8F). These data suggested IgG antibodies mediate VN binding to OC43 infected cells.

## 4. Discussion

Complement (C’) evasion is critical for pathogens to survive this potent host immune response. A gap in knowledge exists about how coronaviruses evade C’-mediated lysis of infected cells. Here, we have used a novel real-time cell viability assay to show that NHS lyses OC43-infected lung cells at a slower rate than PIV5 infected cells. NHS treatment of cells that are co-infected with OC43 and PIV5 results in slow cell killing similar to that of OC43 infected cells alone, suggesting CoV infected cells have a dominant-acting inhibition of C’ effector functions. OC43 infected lung cells recruit C’ inhibitors VN and CLU from NHS and this recruitment requires antibodies. Taken together, we propose a novel role for VN and CLU recruitment by coronavirus infected cells and are associated with delayed C’-mediated death, which has implications for viral pathogenesis and tissue tropism.

C5b-7 insertion into target cell membranes is inhibited by VN, a serum protein that blocks the metastable lipid binding site on C5b-7 [34]. The C5b-7 complex can still incorporate C8 and C9; however, this complex is lytically inactive [38]. VN can also act as a heparin-binding site to inhibit C9 polymerization [39]. CLU is an additional C’ inhibitor present in serum that can act on the terminal step of MAC formation. CLU inhibits MAC assembly by preventing C5b-7 insertion into the target cell membrane and generates soluble inactive complexes [36].

For bacteria, it is well established that some pathogens can express proteins that recruit VN and CLU in order to evade C’-mediated lysis through inhibiting C5b-9 complex formation. Numerous bacteria encode proteins that have been described to recruit VN reviewed in [40], including *Pseudomonas* *aeruginosa*, *Moraxella catarrhalis* (UspA2), *Haemophilus ducreyi* (DsrA), *Haemophilus influenzae* (Hsf and Protein E), *Neisseria meningitidis* (Msf) and *Neisseria gonorrhoeae* [41,42,43,44,45]. In addition, *Pseudomonas aeruginosa*, *Streptococcus pyogenes*, *Staphylococcus* *epidermidis*, *Staphylococcus aureus*, and *Paracoccidioides brasiliensis* can also express proteins that recruit CLU to limit MAC formation [46,47,48,49].

Although most studies on VN and CLU recruitment by pathogens have focused on bacteria, there are descriptions of some viral proteins shown to be associated with VN binding. Human immunodeficiency virus gp120/160 protein [50] and the hepatitis C virus F protein [51] have been shown to bind VN. Conde et al. (2016) have found that dengue virus, West Nile virus, and Zika nonstructural protein 1 (NS1) can bind to and recruit VN to reduce MAC formation and limit C9 polymerization [52]. Additionally, Kuroso et al. (2015) described dengue virus NS1 to interact with CLU [53].

The most striking finding from our work is that VN alone cannot bind to OC43 infected cells, but rather this binding requires a soluble factor contained in serum. Thus, VN alone is not recruited to the surface of infected cells and cannot bind to the OC43 viral glycoproteins expressed on the host cell surface. Depletion studies showed that VN could be recruited to infected cell surfaces even with serum depleted of C’ factors C3, C5, C6 or C7. Instead, our studies indicate a novel VN binding mechanism, whereby IgG antibodies are both necessary and sufficient to allow the deposition of VN on OC43-infected cells. The properties of antibodies (e.g., glycosylation, subtype, dose-dependence, etc.) that facilitate VN and CLU recruitment to cell surfaces is the subject of ongoing studies.

Serum depleted of VN had similar levels of cell killing as compared to NHS (data not shown). A possible explanation for this result is a redundant function of VN and CLU in the inhibition of C’-mediated lysis. CLU depleted serum is not commercially available at this time. A limitation of our study is a deficiency in the direct correlation of VN and CLU recruitment to OC43-infected cells and delayed C’-mediated lysis. Future work will explore VN and CLU double depletion from serum and determine the role of these inhibitors in the kinetics of C’-mediated lysis of OC43-infected cells.

Our findings have implications for the better understanding of mechanisms by which viruses recruit host inhibitors to evade C’-mediated lysis. Future work will address the mechanism of binding of VN and CLU to coronavirus infected cells, as well as unanswered questions such as how widespread this mechanism is among circulating and pathogenic coronaviruses, which viral components are needed for recruitment, investigating individual serum donors and VN and CLU recruitment, and the correlation between anti-coronavirus antibodies from infection versus vaccination in outcomes from infections. Interestingly, pre-existing antibodies to circulating coronaviruses were negatively correlated with SARS-CoV-2 susceptibility and disease progression [54,55]. Future work will determine if VN and CLU are recruited to SARS-CoV-2 infected cells in an antibody-dependent manner and if OC43-specific antibodies can recognize SARS-CoV-2 infected cells and aid in VN and CLU recruitment.

## Figures and Tables

**Figure 1 viruses-14-00029-f001:**
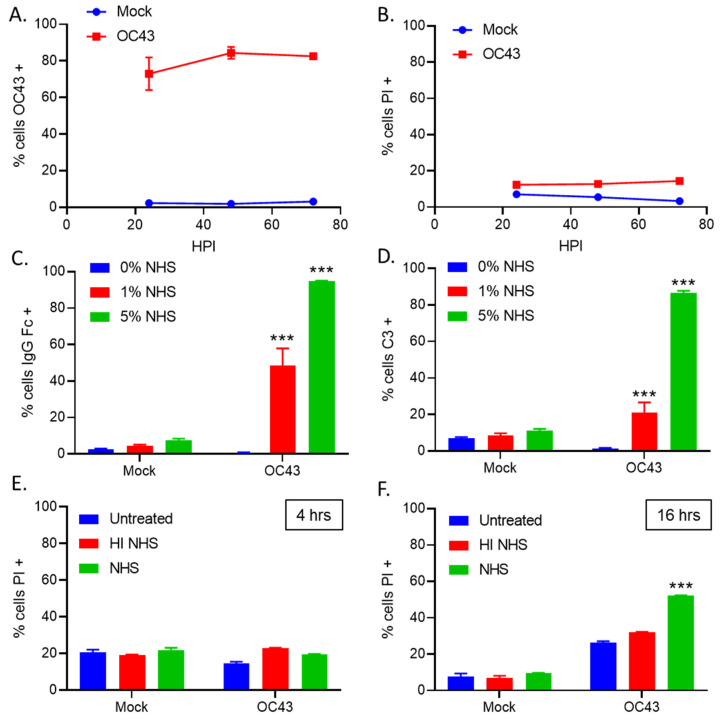
NHS treatment of OC43 infected lung cells results in deposition of antibodies and C3. (**A**,**B**) Human H1299 lung cells were uninfected (mock) or infected with OC43 at an MOI of 5 TCID_50_ units per cell. Cells were stained for intracellular NP expression (**A**) and cell viability was determined by PI staining (**B**) at 24, 48, and 72 hpi. (**C**,**D**) H1299 cells were uninfected (mock) or infected with OC43 at an MOI of 5 TCID_50_ units per cell. Two dpi, cells were either untreated, or treated with 1% or 5% NHS for 1 h at 37 °C. Cells were washed and stained for surface antibody Fc (**C**) or C3 deposition (**D**). (**E**,**F**) H1299 cells were uninfected (mock) or infected with OC43 at an MOI of 5 TCID_50_ units per cell. Two dpi, cells were either untreated or treated with 10% HI NHS or NHS for 4 h (**E**) or 16 h (**F**) at 37 °C. Values in all panels are the mean of three replicates and error bars representing standard deviation, with *** indicating *p*-value of <0.001 comparing HI NHS treatments versus PBS control (**A**), NHS treated OC43 infected versus NHS treated uninfected (mock) samples (**D**,**E**), or NHS treated OC43 infected versus HI NHS treated OC43 infected samples (**F**). Mock denotes uninfected control samples.

**Figure 2 viruses-14-00029-f002:**
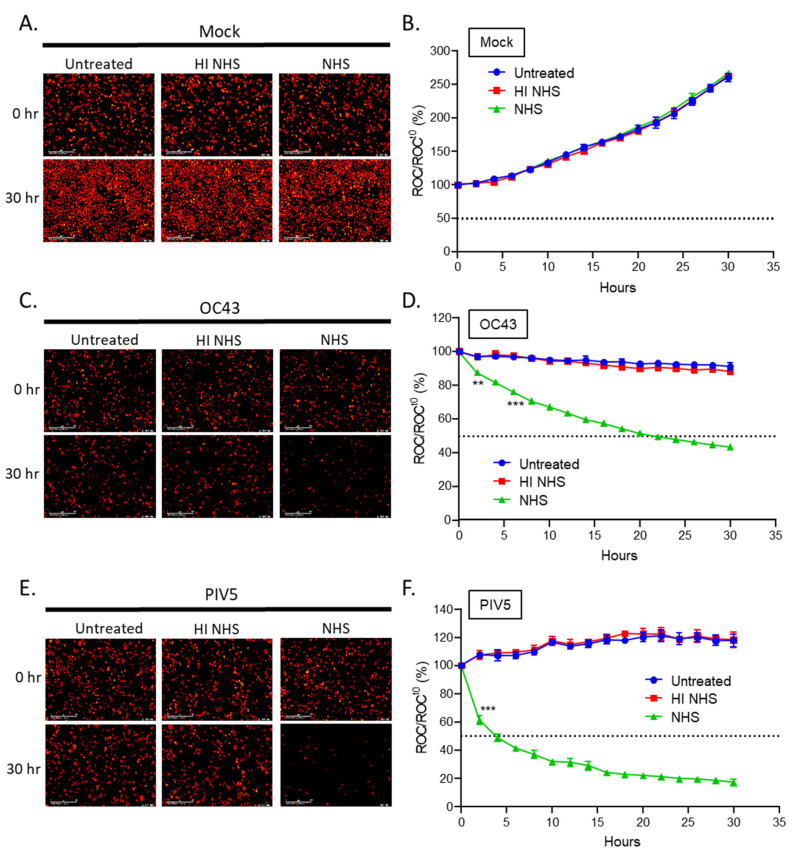
NHS lyses OC43 infected lung cells at a slower rate than PIV5 infected cells using real-time cell viability assays. Human lung H1299-NLR cells expressing nuclear red fluorescent protein were uninfected (mock) (**A**,**B**), or infected with OC43 (**C**,**D**) at an MOI of 5 TCID_50_ units per cell, or PIV5 (**E**,**F**) at an MOI of 5 PFU/cell. Cells were left untreated or were treated with 10% HI NHS or 10% NHS. Bright field phase and red fluorescent images were recorded by real-time imaging using the IncuCyte instrument at intervals of every 2 h over a 30 h. Representative pictures at 0 and 30 h are shown in (**A**,**C**,**E**). Red object count (ROC) per well was calculated and normalized to the value at time zero (ROC^t0^) when treatments were added and expressed as percent of time zero (**B**,**D**,**F**). Values in all panels are the mean of three replicates and error bars representing standard deviation, with ** indicating *p*-value of <0.01 and *** indicating *p*-value of <0.001 comparing NHS treated versus HI NHS treated infected samples. Mock denotes uninfected control samples.

**Figure 3 viruses-14-00029-f003:**
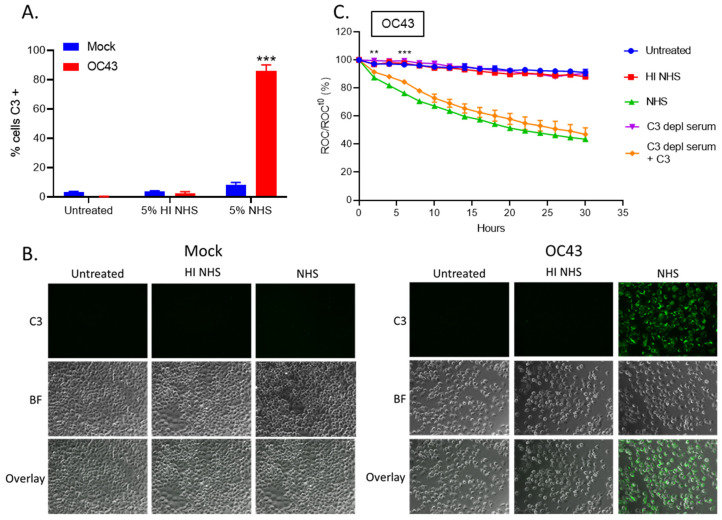
C3 is deposited on OC43 infected lung cells and is required for NHS-mediated cell killing. (**A**,**B**) Uninfected (mock) or OC43 infected H1299 cells were either untreated, HI NHS, or NHS treated for 1 h at 37 °C. After incubation, cells were washed and stained for surface C3 deposition and quantified by flow cytometry (**A**) or by immunofluorescence (**B**). (**C**) Human lung H1299-NLR cells were uninfected (mock) or OC43 infected, and then either left untreated or treated with 10% HI NHS, 10% NHS, 10% C3-depleted (depl) serum, or 10% C3-depl serum supplemented with physiological levels of C3. Red fluorescence was acquired at intervals of every 2 h over 30 h by real-time imaging using the IncuCyte instrument. ROC per well was calculated and normalized to the value at time zero (ROC^t0^) when treatments were added and expressed as percent of time zero. Values in all panels are the mean of three replicates and error bars representing standard deviation, with ** indicating *p*-value of <0.01 and *** indicating *p*-value of <0.001 comparing NHS treated versus HI NHS treated infected samples (**A**) or NHS versus C3 depl serum treated infected samples (**C**). Mock denotes uninfected control samples.

**Figure 4 viruses-14-00029-f004:**
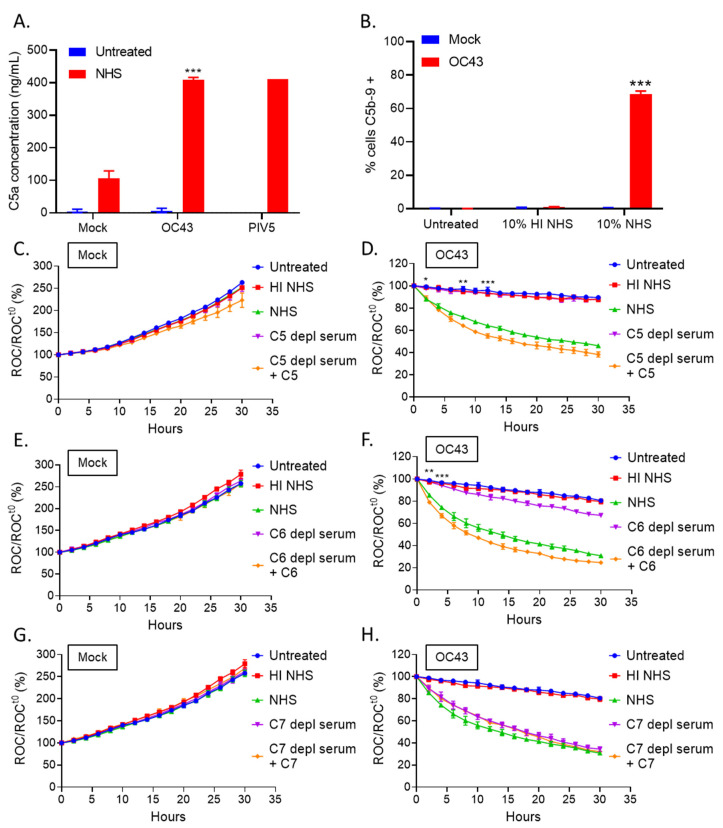
C5 and C6 are required for C’-mediated lysis of OC43 infected lung cells. (**A**) Uninfected (mock), OC43, or PIV5 infected H1299 cells were either left untreated, or treated with NHS for 4 h at 37 °C. Supernatants were collected and analyzed for C5a concentrations via ELISA. (**B**) Uninfected (mock) or OC43 infected H1299 cells were either left untreated, or treated with HI NHS, or NHS treated for 1 h. After incubation, cells were washed and stained for surface C5b-9 and quantified by flow cytometry. (**C**–**H**) H1299-NLR cells were uninfected (mock) (**C**,**E**,**G**) or infected with OC43 (**D**,**F**,**H**) at an MOI of 5 TCID_50_ units per cell and then were either left untreated or treated with 10% HI NHS, 10% NHS, 10% C5-depl serum, or 10% C5-depl serum supplemented with physiological levels of C5 (**C**,**D**). Brightfield phase and red fluorescent images were acquired by real-time imaging using the IncuCyte instrument at 2 h intervals over 30 h. ROC per well was calculated and normalized to the value at time zero (ROC^t0^) when treatments were added and expressed as percent of time zero. For panels (**E**,**F**), 10% C6-depl serum or 10% C6-depl serum supplemented with physiological levels of C6 were tested as indicated. For panels (**G**,**H**), C7-depl serum or 10% C7-depl serum supplemented with physiological levels of C7 were tested as indicated. Values in all panels are the mean of three replicates and error bars representing standard deviation, with * indicating *p*-value of <0.05, ** indicating *p*-value of <0.01 and *** indicating *p*-value of <0.001 comparing NHS treated OC43 infected versus NHS treated uninfected (mock) samples (**A**), NHS treated OC43 infected versus HI NHS treated OC43 infected samples (**B**), or NHS versus depl sera treated infected samples (**D**,**F**). Mock denotes uninfected control samples.

**Figure 5 viruses-14-00029-f005:**
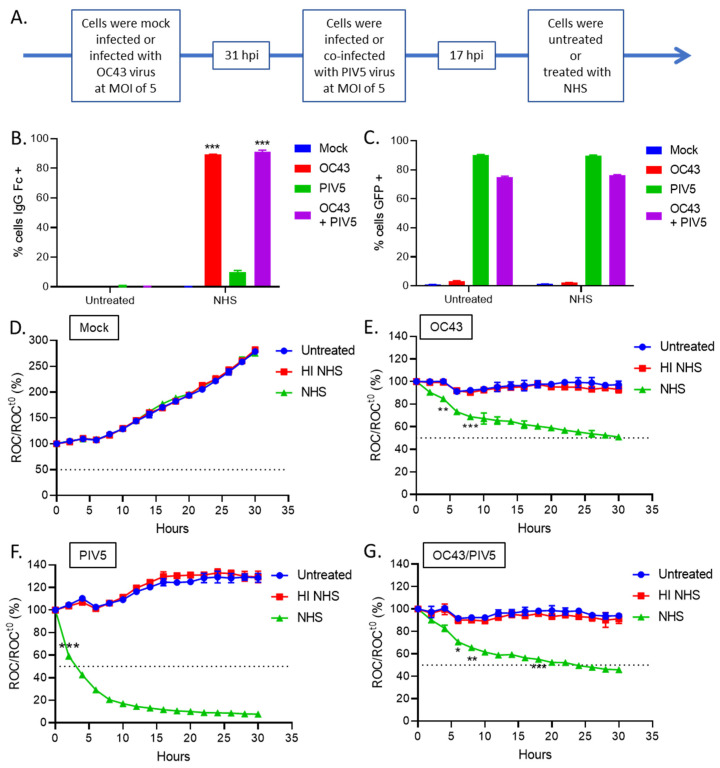
NHS treatment of cells that are co-infected with OC43 and PIV5 results in slow cell killing similar to that OC43 infected cells alone. (**A**–**C**) H1299 cells that were uninfected (mock), or infected with OC43 or PIV5 alone, or co-infected with OC43 and PIV5 cells were either left untreated or treated with NHS for 1 h. Cells were then stained with anti-Fc antibody and assayed for surface Fc (**B**) and GFP expression (**C**) by flow cytometry. (**D**–**G**) H1299-NLR cells were uninfected (mock) (**D**), infected alone with OC43 (**E**) or PIV5 (**F**), or co-infected with OC43 and PIV5 (**G**). Cells were either left untreated or treated with 10% HI NHS, or 10% NHS. Red fluorescent images were acquired by real-time imaging using the IncuCyte instrument at 2 h intervals over 30 h. ROC per well was calculated and scans were normalized to the value at time zero (ROC^t0^) when treatments were added and expressed as percent of time zero. Values in all panels are the mean of three replicates and error bars representing standard deviation, with * indicating *p*-value of <0.05, ** indicating *p*-value of <0.01 and *** indicating *p*-value of <0.001 comparing NHS treated OC43 or OC43 +PIV5 infected samples versus NHS treated uninfected (mock) samples (**B**), or NHS treated infected samples versus HI NHS treated infected samples (**E**–**G**). Mock denotes uninfected control samples.

**Figure 6 viruses-14-00029-f006:**
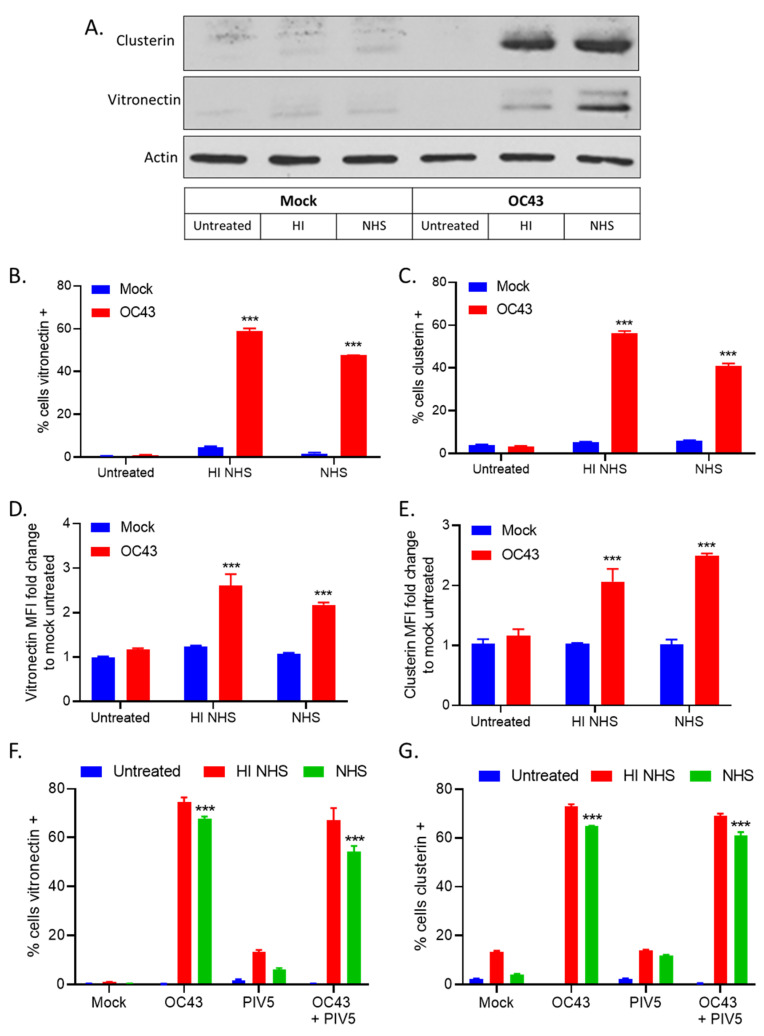
OC43 infected lung cells recruited VN and CLU from NHS. (**A**) Uninfected (mock) or OC43 infected H1299 cells were either left untreated, or treated with HI NHS, or NHS for 1 h. Cells were washed and lysates were probed for VN and CLU. (**B**–**E**) Uninfected (mock) or OC43 infected cells were left untreated or treated with 10% HI NHS, or 10% NHS for 1 h before staining for surface binding of VN (**B**), or CLU (**C**). Data are expressed as percentage that are antibody positive (**B**,**C**) or MFI (**D**,**E**). (**F**,**G**) Cells that were uninfected (mock), or singly infected with OC43 or PIV5, or co-infected with PIV5 and OC43 were either left untreated, or treated with HI NHS, or NHS for 1 h and stained for surface binding of VN (**F**) or CLU (**G**). Values in all panels are the mean of three replicates and error bars representing standard deviation, with *** indicating *p*-value of <0.001 comparing HI NHS or NHS treated infected cells versus HI NHS or NHS treated uninfected (mock) samples. Mock denotes uninfected control samples.

**Figure 7 viruses-14-00029-f007:**
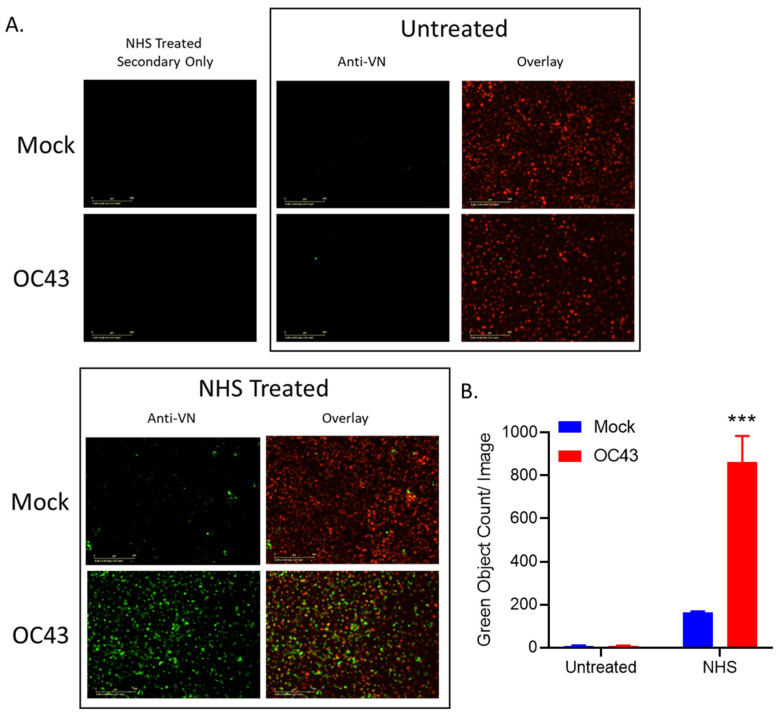
VN from serum was recruited to the surface of OC43 infected lung cells. (**A**,**B**) Uninfected (mock) or OC43 infected H1299-NLR cells were either untreated or treated with NHS for 2 h. Cells were washed, fixed, and stained for surface VN as detected by immunofluorescence. Cells were then imaged on the IncuCyte under 20× objective (**A**) and VN positive stained cell counts were determined per image (**B**). Red corresponds to the cell nuclei expressing red fluorescent protein and green corresponds to VN staining. Values are the mean of three replicates and error bars representing standard deviation, with *** indicating *p*-value of <0.001 comparing NHS treated uninfected (mock) cells versus NHS treated OC43 infected samples. Mock denotes uninfected control samples.

**Figure 8 viruses-14-00029-f008:**
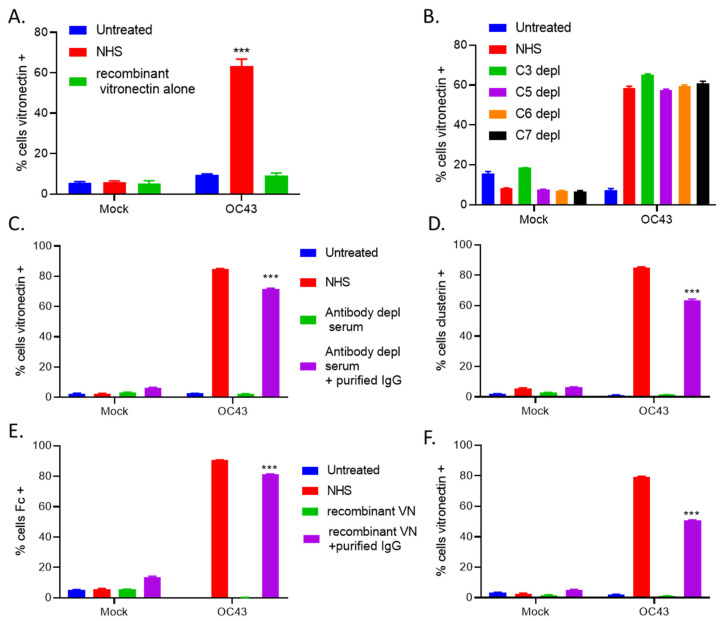
VN was bound to OC43 infected lung cells independently of key C’ pathway components and VN and CLU recruitment required antibodies from NHS. (**A**) Uninfected (mock) or OC43 infected H1299 cells were either untreated or treated with NHS or physiological levels of recombinant VN for 1 h at 37 °C. Cells were washed and probed for VN surface staining. Values are the mean of three replicates and error bars representing standard deviation, with *** indicating *p*-value of < 0.001 comparing NHS treated versus recombinant VN alone treated infected samples. (**B**) Uninfected (mock) or OC43 infected cells were left untreated or treated with 10% NHS, or 10% C3, C5, C6, or C7 depleted (depl) serum for 1 h at 37 °C. Cells were washed and probed for VN (**B**) surface staining. (**C**,**D**) Uninfected (mock) or OC43 infected cells were untreated or treated with 10% NHS, or 10% antibody depl serum, or 10% antibody depl serum plus physiological levels of purified IgG antibodies from human serum for 1 h at 37 °C. Cells were washed and probed for VN (**C**), or CLU (**D**) surface staining. *** indicated *p*-value of <0.001 comparing antibody depl serum treated infected samples versus antibody depl serum plus purified IgG treated infected samples. (**E**,**F**) Uninfected (mock) or OC43 infected cells were untreated or treated with 10% NHS, or treated with recombinant VN alone, or recombinant VN plus purified IgG antibodies from human serum for 1 h at 37 °C. All treatments were performed using physiological serum levels of recombinant VN (200 μg/mL) and purified IgG (7 mg/mL). Cells were washed and probed for antibody Fc (**E**), or VN surface staining (**F**). Values in all panels are the mean of three replicates and error bars representing standard deviation, *** indicating *p*-value of <0.001 comparing recombinant VN treated infected cells versus recombinant VN plus purified IgG treated infected samples. Mock denotes uninfected control samples.

## Data Availability

Data availability is at request.

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
