# Peer review of "Complement Inhibitors Vitronectin and Clusterin Are Recruited from Human Serum to the Surface of Coronavirus OC43-Infected Lung Cells through Antibody-Dependent Mechanisms"

_viruses, 2021, doi:10.3390/v14010029_

Round 1

Reviewer 1 Report

Study by Fox and Parks address very interesting and important question on mechanisms of antibody-dependent complement-mediated killing of virus-infected cells strategies that viruses used to evade these immune reactions. Study provides comprehensive evidence of IgG-dependent recruitment of vitronectin and clusterin to the plasma membrane of OC43-infected cells and authors suggest that this recruitment delays complement-mediated cell killing. Such delay can be detrimental even for immunocompetent host since it will widen window of opportunity for virus allowing it to produce more progeny and thus outcompete immune system. The study performed at high level with sound methodology and results are interpreted correctly. However, there are several major and few minor concerns that have to be addressed.

Major comments:

  1. Title should include OC43 otherwise it is misleading, especially in current situation with COVID-19 pandemic when mass media excessively freely interpret scientific findings.

The word “correlated” should also be avoided in title. Authors did not perform statistical analysis on whether antibody-dependent recruitment of VN and CLU correlates with delayed cell killing. In fact, authors show only that lysis of OC43-ionfected cells is slower than PIV5-infected cells and that VN and CLU binding to the surface of infected cell is IgG-dependent. There is no direct experiment showing if IgG-depletion or depletion of OC43-specific IgG would cause similar effect or effect would be abrogated/altered.

  1. Introduction is inappropriately general, generic and largely adapted from Li et al., (Virology, 2016) from the same laboratory. It lacks hypothesis formulation as well as information on vitronectin and clusterin. Discussion actually contains a few paragraphs describing CLU and VN and their role in inhibiting complement reaction during bacterial and viral infections. This information is more appropriate in Introduction.
  2. The major hypothesis that set to be tested in this study is that OC43 uses different strategy compared to PIV5 to slow down complement-dependent cell lysis. Normal human serum contains OC43-specific antibodies but not PIV5 antibodies since PIV5 is a simian virus. This should be clearly stated in the text. In their prior study (Mayer et al., Virology 2014), same laboratory showed that serum form naïve animals can potently neutralize PIV5 via complement-dependent mechanism suggesting that mechanisms of cell killing, and lysis is antibody-independent.
  3. Figure 3 and data description in the text: Presented data convincingly demonstrate that slow cell lysis of OC43-infected cells is not due to impaired opsonization with C3 component. However, it is not clear whether C3 component is deposited equally well on OC43- and PIV5-infected cells, especially given that authors mention that “PIV5 to be a potent and rapid activator of C’ pathways”. In Johnson et al., (Virology, 2008) article by the same group, authors show that PIV5 virions can bind C3 and that C3 is required for virus neutralization by normal human serum. Therefore, C3 is to be expected binding to PIV5-infected cells. Inclusion of quantitative comparison of C3 deposition on the surface of OC43-infected cells and PIV5-infected cells can be suggested.

Another issue is the statement that PIV5 is potent and rapid activator of complement pathways. How were the potency and speed of complement activation measured? I could not find any prior work which would comparatively show that PIV5 activates complement faster or more potently than other viruses. It seems that this statement is solely based on the results of the current study and thus should be moved into Discussion and properly addressed.

  1. It is not clear for how long cells were infected with PIV5 in coinfection experiments and after what time normal human serum was added. Additionally, did authors try to infect cells first with PIV5 and then co-infect with OC43? Such experiment would help clarifying the effect of OC43 infection on cell response to PIV5 infection.
  2. In section 3.4, authors should clarify that normal human serum should not contain antibodies to PIV5 and that is why PIV5-infected cells lack IgG opsonization. It also should be noted that according to the prior work of the same laboratory PIV5 activates complement via alternative pathway and serum collected form PIV-5-naïve animals can neutralize PIV5 via complement-dependent mechanism. This is important moment in understanding the rationale of coinfection experiments and subsequent result interpretation.
  3. Taken together these results suggested VN and CLU can bind to OC43 infected cells and this inhibition can overcome potent C’ activation induced by PIV5 infection”. There is no evidence or quantitative information on potency of C’ activation by PIV5 was provided. It is clear that PIV5-induced mechanism kills cells much faster than OC43 but it is different, antibody-independent mechanism as such comparison of virus is not appropriate but comparison of mechanisms can be. The difference in the mechanisms not in viruses and that should be emphasized.
  4. Author state that the following: “Taken together, we propose a novel role for VN and CLU recruitment by coronavirus infected cells to delay C’-mediated death, which have implications for viral pathogenesis and tissue tropism”. This statement in my opinion is not directly supported by the presented data. Although IgG was shown critical for the VN binding to the surface of OC43-infected cells and cell killing is also relies on IgG opsonization, authors do not demonstrate a direct link between VN recruitment and delayed cell lysis. Experiments with VN-depleted serum and complement-mediated cell killing, similar to those presented on Figure 1, would help to clarify this concern. Also, experiments on VN and CLU-mediated delay of OC43/PIV5 coinfected cells would benefit this study so they can shed some light on the role of VN and CLU in the delayed killing of OC43-infected cells.
  5. It is also not clear if VN and CLU role is complementary, independent or interdependent. Authors state in the last paragraph of Results that “Serum depleted of VN had similar levels of cell killing as compared to NHS (data not shown). A possible explanation for this result is a redundant function of VN and CLU in inhibition of C’-mediated lysis. CLU depleted serum is not commerically available at this time”. This is actually a key point of the study – authors propose a new mechanism of slow cell lysis of OC43-infected cells because of recruitment of VN and CLU to the surface of infected cells. Although logical, provided evidence is indirect and based on the known VN and CLU functions, their IgG-dependent recruitment to the plasma membrane of infected cells and that OC43-infected cells are killed at a slower rate compared to PIV5-infected. Since antibodies against CLU are commercially available, serum depletion can be easily performed in the lab hence experiment with serum depleted with both VN and CLU is recommended. VN-, CLU- and VN/CLU-depleted serum should be also analyzed in the kinetic of OC43-infected cell lysis in IncuCyte-assisted experiments. This will provide direct evidence on the involvement of VN and CLU in this process.
  6. Discussion of study limitations is lacking. For example, can these effects be reproduced with individual serum samples? Are IgG-mediated effects dose-dependent?

Minor comments:

  1. Figure 1 legend, last line: panel F not panel G should be mentioned.
  2. Page 7, bottom paragraph: sequence of references should be corrected to read [25, 27, 31-32]
  3. Figure 5 would benefit from including dashed line showing 50% killing.

Reviewer 2 Report

Overall, the experimental design was systematically performed. The authors are providing valuable evidence on the Complement-Mediated Lysis of Lung Cells promoted by Human Coronavirus.

The following are suggestions for improvement:

Minor comments

  1. Authors refers in the manuscript as “mock infected” and make a bit confusing. I suggest replacing the term “mock infected” by uninfected or uninfected (mock).
  2. For figure 2B. to make it easy the understanding, please include on the picture the word “Uninfected”
  3. For figure 2D. to make it easy the understanding, please include on the picture the word “OC43”
  4. For figure 2D. To make it easy the understanding, please include on the picture the word “PIV5”
  5. Figure 3C, to make it easy the understanding, please include on the picture the word “OC43”
  6. For figure 4C, E and G. to make it easy the understanding, please include on the picture the word “Uninfected”
  7. For figure 4D, F and H. to make it easy the understanding, please include on the picture the word “OC43”
  8. Figure 5. It’s hard to follow the rationale, to make it easy the understanding, please include a schematic for the experiment.
  9. Figure 5B. Authors should also analyse the number or percentage of infected cells with OC43.

Round 2

Reviewer 1 Report

All suggestions and critique was adequately addressed.